# Reliability of Domestic Gas Flow Sensors with Hydrogen Admixtures

**DOI:** 10.3390/s24051455

**Published:** 2024-02-23

**Authors:** Giorgio Ficco, Marco Dell’Isola, Giorgio Graditi, Giulia Monteleone, Paola Gislon, Pawel Kulaga, Jacek Jaworski

**Affiliations:** 1Department of Civil and Mechanical Engineering, University of Cassino and Southern Lazio, 03043 Cassino, Italy; dellisola@unicas.it; 2Department of Energy Technologies and Renewable Sources, Agenzia Nazionale per le Nuove Tecnologie, l’Energia e lo Sviluppo Sostenibile (ENEA), Research Center Casaccia, 00123 Rome, Italy; giorgio.graditi@enea.it (G.G.); giulia.monteleone@enea.it (G.M.); paola.gislon@enea.it (P.G.); 3Oil and Gas Institute—National Research Institute, ul. Lubicz 25a, 31-503 Kraków, Poland; kulaga@inig.pl (P.K.); jaworski@inig.pl (J.J.)

**Keywords:** smart meter, in-field verification, ultrasonic, thermal mass, diaphragm, hydrogen blending, hydrogen admixture

## Abstract

Static flow sensors (e.g., thermal gas micro electro-mechanical sensors—MEMS—and ultrasonic time of flight) are becoming the prevailing technology for domestic gas metering and billing since they show advantages in respect to the traditional volumetric ones. However, they are expected to be influenced in-service by changes in gas composition, which in the future could be more frequent due to the spread of hydrogen admixtures in gas networks. In this paper, the authors present the results of an experimental campaign aimed at analyzing the in-service reliability of both static and volumetric gas meters with different hydrogen admixtures. The results show that the accuracy of volumetric and ultrasonic meters is always within the admitted limits for subsequent verification and even within those narrower of the initial verification. On the other hand, the accuracy of the first generation of thermal mass gas flow sensors is within the limits of the verification only when the hydrogen admixture is below 2%vol. At higher hydrogen content, in fact, the absolute weighted mean error ranges between 3.5% (with 5%vol of hydrogen) and 15.8% (with 10%vol of hydrogen).

## 1. Introduction

Measurement and billing of natural gas consumption involves several technical, metrological and consumer protection issues. At the same time, individual smart metering is considered an effective tool to improve energy savings in the residential sector, which accounts for 30% of global energy consumption and 26% of global energy-related emissions [1]. Furthermore, the diffusion of smart metering technologies is expected to lower the so-called ‘Delta In-Out’, that is the unbalance resulting from the unavoidable differences between the gas entering and leaving the distribution networks [2].

Together with water meters, gas meters are the most traditional within the utility meters included in the MID, the Measuring Instruments Directive [3]. According to the latter, gas meters must comply with strict essential metrological requirements (e.g., errors, environmental and mechanical classes), even in-service. In the EU, the certification and verification of utility meters for the legal metrology use ensure accuracy and fairness in gas billing. In this context, gas meter manufacturers must have their meter models subjected to type examination by a Notified Body accredited by the relevant national authority. After type examination, each gas meter produced must undergo an initial verification to ensure that it meets the required accuracy limits and complies with legal metrology requirements.

Regardless of the measuring principle of the flow sensor, utility meters are subject to a natural drift over time. Therefore, once put into service, subsequent periodic verifications apply, involving regular re-inspection and check of gas meters with the aim of ensuring that they remain accurate and reliable during their service life. In this regard, a twofold approach is adopted by legal metrology authorities: (i) meters are removed from service and replaced after a certain time period of validity of the legal seals (e.g., in Italy 15 years); (ii) periodic verification is carried out at fixed frequency. In some member states, a statistical control approach is applied to lots of meters [4]. Finally, since EU member states must ensure that consumers have the right to accurate metering and billing for gas consumption, consumers may request verification of their gas meters of any size if they suspect inaccuracies and have any discrepancies addressed.

While the MID sets, at EU level, the legal rules for type approval, production and initial verification of domestic gas meters, single EU member states may have their specific national regulations for the subsequent verification. In fact, it must be ensured that the accuracy and fairness of gas billing for consumers is maintained over time. Among these requirements, the frequency of subsequent verifications is set by the relevant national regulations and may vary from one member state to another. Both MID and national regulations specify the acceptable tolerances and accuracy classes for gas meters, determining how much a gas meter’s reading can deviate from the actual gas consumption. According to the applicable technical recommendation [5], gas meters are classified in three accuracy classes (i.e., class 0.5, 1 and 1.5) for which a maximum permissible error (MPE) of 0.5%, 1.0% and 1.5% is set in the range between the transition ($Q_{t}$) and maximum ($Q_{max}$) flow rate, whereas in the range between the minimum ($Q_{min}$) and transition ($Q_{t}$) the MPE is equal to 1.0%, 2.0% and 3.0%. The MID directive, however, only provides for classes 1 and 1.5, stating also that class 1.5 applies for the residential sector.

In Italy, subsequent verification of gas meters is regulated by the ministry decree 97/2017 [6], according to which gas meters with $Q_{max}$ above 10 m^3^/h are verified with a frequency ranging between 10 years (turbine and rotary type) and 16 years (diaphragm type), while for those of other technologies (e.g., ultrasonic and thermal mass) a frequency of 8 years is adopted. On the technical hand, the operational procedure is established in the Ministry Directive of 26 July 2023 in continuity with of the UNI 11600 series standards [7].

In principle, subsequent verification of gas meters should be provided at the installation site aiming at guaranteeing the actual installation condition of use of the meter. However, it is almost always conducted in the laboratory aiming at guaranteeing safe and reliable test conditions and to fulfil the requested uncertainty values, also lowering the related costs. Obviously, the field installation conditions cannot be fully reproduced in the laboratory where the so-called rated conditions apply, and this can lead to possible deviation of the results from the conditions of use. Nevertheless, field conditions are also tested during type approval tests (e.g., flow disturbance tests), even if all field contributions cannot be completely addressed.

Nowadays, blending green hydrogen (e.g., that produced by electrolysis from renewable energy sources) in the Natural Gas (NG) infrastructures is assuming increasing interest within Distribution System Operators (DSO), aiming at reaching the decarbonization objectives. However, this practice influences the quality of the gas supplied and particularly its thermophysical properties such as density, specific heat and gross calorific value [8,9,10,11]. Therefore, consequences on the appliances at end-user’s level are likely [12], as well as potential inaccuracies of the domestic gas meters in distribution networks [13]. In particular, from the analysis of the evidence so far available in the scientific literature for brand-new gas meters, it can be highlighted that:Diaphragm gas meters show no significant metrological deviations in terms of average drift of errors using natural gas and hydrogen admixture (H2NG) up to 15%vol [14,15];Almost stable metrological performance was found for domestic G4 ultrasonic gas flow sensors tested with H2NG mixtures with hydrogen admixture up to 10%vol [9];Thermal mass gas flow sensors show significant errors (i.e., much higher than the corresponding MPEs) when tested with gas with hydrogen content of 10%vol and 15%vol and occasionally, even with lower concentrations (2, 4 and 5%vol) [16].

It can be affirmed that the good reliability found in the literature of diaphragm and ultrasonic gas flow sensors with H2NG was almost expected, as well as the decay of the thermal mass gas flow sensors of the first generation due to the measuring principle which is based on the thermophysical properties of the gas being measured [17,18,19].

On the other hand, the scientific literature concerning the results of periodic verification of domestic gas meters is almost lacking and always related to tests in air at laboratory conditions. In [20] the results of a wide experimental analysis conducted in Italy on diaphragm gas meters show that the percentage of very old diaphragm gas meters conforming subsequent verification limits is about 52%, whereas for those with service life below 15 years this percentage grows up to 65%. Statistics available for the UK market [21] show that 75.4% of disputed domestic meters in 2014 were found to be accurate, together with 74.1% of disputed commercial meters (i.e., with maximum flow rate between 6 m^3^/h and 160 m^3^/h). In [22] the statistical analysis of the outcomes of subsequent verification of 1582 G25 industrial thermal mass meters installed in 2013 and removed from the field in 2021 are presented. The authors found that about 97% of the sample complies with the limits of in-service verification and that about 80% still complies with the initial verification accuracy provisions. Finally, in [23] four diaphragm (which were previously used for laboratory tests) and three ultrasonic (which were removed from the field) gas meters have been tested with pure methane and with a blending of methane and 20%vol hydrogen. The authors demonstrated that:The diaphragm gas meters did not show any significant change in accuracy;The ultrasonic gas meters showed more spread results and the tendency to underestimate with the hydrogen blending, especially at lower flow rates.

In this paper, the authors present the results of an experimental campaign carried out in the laboratory on three domestic G4 gas meters with different flow sensors previously removed from service. Tests were carried out in air and natural gas of H family [24] and with the same gas with hydrogen admixture of 2, 5, 10 and 23%vol. To the author best knowledge, this is the first paper devoted to the analysis of the effect of hydrogen admixture on the reliability of domestic gas meters removed from service. The tested gas meters belong to the three different flow sensor types (i.e., diaphragm, ultrasonic and thermal mass) representing almost the totality of household gas meters. Furthermore, they have been submitted to tests with the same gases and at the same laboratory conditions, thus enhancing the effectiveness of the comparison. Also, the potential deviation between subsequent verification outcomes when tests are performed with air and real natural gases (i.e., not pure methane) has been investigated. The obtained results could be useful to different stakeholders in the field of domestic gas metering. As for example, specific plans of subsequent verification/replacement of installed meters could be developed by DSOs and more appropriate verification intervals tailored on the measuring principle could be set by National Authorities. On the other hand, the main limitation of this study is represented by the limited sample of meters tested and that only laboratory tests have been developed and discussed. This implies that some of the measured deviations could be even larger due to the unavoidable effects of additional influence related to the installation conditions and to the actual operating conditions.

## 2. Materials and Methods

In modern city networks, smart metering for domestic consumption of natural gas is mainly performed through diaphragm, ultrasonic and thermal mass gas meters, which technical requirements are regulated through the related harmonized standards [25,26,27]. The traditional measurement technology used for the residential sector is the volumetric one. Among the so-called static meters (i.e., those without measuring moving parts) ultrasonic and thermal mass flow sensors are the most spread.

Volumetric gas meters, also known as Positive Displacement meters, directly measure the volume passed by isolating volumes of gas that alternatively fill and empty compartments with a cyclic nominal volume. In the residential sector, diaphragm type are the most spread volumetric gas meters. Accuracy of diaphragm gas meters is related to potential leakages and geometrical errors as well as to the unavoidable wear of materials over time (especially that of the diaphragms, which before 1990 were made up of animal leather). This type of meters shows good reliability and low costs, whereas the presence of moving parts leads to not negligible pressure drop and unavoidable wear. Aiming at complying with the current smart metering requirements, hybrid diaphragm gas meters (in which the mechanical output is converted electronically and equipped with transmission devices) are currently produced and installed.

Domestic ultrasonic sensors use the time-of-flight principle basing on the measurement of the transit time spent by an acoustic wave in passing from an emitter to a receiver in the gas stream. At domestic level, a single-path configuration and a measuring section where the acoustic wave is eventually reflected one or more times are used. On the metrological hand, ultrasonic gas flow sensors present good performance and not particularly high costs. However, the effect of contaminants could be significant in city networks. Ultrasonic flow sensors for domestic gas meters are designed to work with speed of sound in the gas stream between 300 and 475 m/s [26]. Their use is almost exclusively for gases of the second family of EN 437 [24] and, particularly for groups H and E with high methane content, although the use with other gases is possible. Obviously, the hydrogen admixture in natural gas influences the speed of sound, which can be calculated through AGA 10 method [28]. The speed of sound in the medium increases as the hydrogen content in the mixture increases and this could lead to inaccuracies. As for example, the admixture of 17%vol of hydrogen at T = 15 °C in pure methane leads the speed of sound to be approximately 475.5 m/s, above the quoted limit of 475 m/s.

Thermal mass sensors are based on the rate of heat transfer between a micro electro-mechanical sensor (MEMS) and the gas stream. The use of thermal mass flow meters in natural gas distribution began in 2003 and they currently represent one of the most promising technologies for gas measurements in the domestic field. The heat transfer is measured through two temperature sensors symmetrically installed upstream and downstream of a micro heater. At zero-flow, the two temperature sensors measure the same temperature, whereas the temperature symmetry is disturbed when a gas stream flows. Thus, a temperature difference occurs between the two sensors, which is proportional to the mass flow rate of the gas. The heat transfer depends on the gas mass flow rate, as the main effect, but also on the gas composition, which influences its thermo-physical properties, like thermal conductivity and diffusivity. Therefore, the gas identification represents the main potential limit of this technology, and the more recent developments of this technology rely on additional functionalities devoted to recognizing the quality of the gas flowing. Furthermore, the measured flow rate has automatic temperature and pressure compensation, and direct measurement of the calorific value is also possible. These latter characteristics make them ideal for the current demand for direct measurements in gas energy.

### 2.1. Subsequent Verification of Gas Meters

In Italy subsequent verification of domestic gas meters is performed according to [6] and national standards [7]. The relative error ($E_{MUT}$) of the meter under test (MUT) is calculated as per Equation (1), in which $I_{MUT}$ is the indication of the meter and $I_{ref}$ is that of the reference master meter or reference measuring system. Verification is passed if $E_{MUT}\leq MPE$ at all verification points.
(1)$$E_{MUT}=\frac{I_{MUT}-I_{ref}}{I_{ref}}\times 100$$

*MPEs* for new instruments are set in instrument specific legislation, the primary source being the EU directives, such as non-automatic weighing instruments (NAWI) and MID, whereas for in-service MPEs different approaches are set. For example, for NAWIs, the directive itself prescribes that the in-service MPEs shall be twice those for new instruments. In the MID, the in-service MPEs are left completely to the discretion of the national authorities. Traditionally, for many instrument categories including gas meters, it has been usual to set the in-service MPEs as twice those for new instruments [29], such as in Italy and in Poland. In Table 1, the MPEs in subsequent verification of gas meters are thus reported. It is worth highlighting that in Italy, according to [6], for the meter types equipped with conversion devices as a function of the temperature which indicate only the converted volume, the MPE for new instruments is increased by 0.5% and this specifically applies to thermal mass gas meters. Furthermore, the fluid to be used for the subsequent verification is included among those of the gas families for which the meter is suitable. However, if the verification is carried out in a laboratory, the use of air is also possible, unless explicitly excluded in the type approval documents of the meter.

In [6] it is also mandatory that the master meter used to carry out the periodic verification must not be affected by an error $E_{MM}$ greater than 1/3 of the MPE of the MUT and that the related expanded uncertainty of the master meter $U_{MM}$ must not exceed 1/3 of the $E_{MM}$ itself (i.e., 1/9 of MPE of the MUT). Such requirement, which is a general legal metrology provision [29], is a very challenging condition, especially in the field, since it leads to uncertainty of 0.33% (0.66%) in the range $Q_{t}\leq Q<Q_{max}$ ($Q_{min}\leq Q<Q_{t}$).

As is well known, the MUT resolution plays a significant role in determining the verification uncertainty. In fact, since the indication of the MUT is obtained as the difference between the final and the initial readings, then $u_{R,final}=u_{R,initial}=R/(2\sqrt{3})=R/\sqrt{12}$, where R is the MUT resolution. By considering a rectangular probability distribution, it follows that the combined uncertainty is $u_{R}$ = $\sqrt{{2R}^{2}/12}=R/\sqrt{6}$ and that the relative combined uncertainty for the MUT resolution is $u_{R,\%}=u_{R}/Q$ (being Q an integer number of digits, i.e., Q=n R, and $U_{R,\%}<MPE/3$). Therefore, a minimum test quantity (MTQ) calculated as per Equation (2) should be adopted.
(2)$$MTQ=\frac{2\;\;R\;}{\sqrt{6}\;\;\frac{MPE}{3}}$$

### 2.2. The Experimental Campaign

The authors specifically designed an experimental campaign aimed at evaluating the reliability of gas flow sensors in-service, in particular:Tests in air were first performed at the 2i Rete Gas Laboratory of Cremona with a sonic nozzle test bench capable to guarantee expanded uncertainty (k = 2), excluding the MUT contribution, within 0.28% (0.44%) in the range $Q_{t}$ − $Q_{max}$ ($Q_{min}$ − $Q_{t}$);Tests with H family gas and H2NG have been performed at the Department of Flow Metrology of the Oil and Gas National Research Institute in Kraków (PL). The expanded uncertainty (k = 2), excluding the MUT contribution, is within 0.39% (0.58%) in the range $Q_{t}$ − $Q_{max}$ ($Q_{min}$ − $Q_{t}$).

In both cases the expanded uncertainty is below the admitted limit of 1/3 MPE. Tests with H family gas and H2NG were carried out by using the standing start-stop volumetric method. Figure 1 shows the sketch of the test bench.

Gas mixtures are prepared in a modular mixer using calibrated mass regulators and control software. Source gases from pressure cylinders (ethane, methane, hydrogen, nitrogen) feed the modular mixer and the software allows control of the required mixture composition. To validate the given compositions of gas mixtures, the laboratory provides gas chromatographic measurements on gas samples taken from the test bench. The mixture ratio is then punctually monitored via a continuous measurement using the mass regulators of the modular mixer.

Downstream to a first pressure regulator, two separate loops are available by alternatively closing dedicated shut-off valves: (i) in the high flow rate loop, the gas flow is controlled by an intrinsically safe side-channel blower, where speed is regulated through an inverter; (ii) in the low flow rate loop, the flow is controlled by a fine adjustable valve downstream to the MUTs. A burner evacuates the gas at the exit of the bench.

Gas temperature is measured using four Pt100 temperature sensors (manufacturer Termoaparatura Wrocław, Zębice, Poland) with EN 60751 accuracy class A (one in each test loop and two downstream and upstream to the MUTs). The test bench is also equipped with six pressure transmitters: (i) three gauge pressure transmitters type PC-28 PD Exi (manufacturer Aplisens S.A., Warsaw, Poland) for the measurement of gauge pressure in each test loop and in the MUT and (ii) three differential pressure transmitters type APR-2000GDP (manufacturer Aplisens S.A., Warsaw, Poland) for the measurement of differential pressure in each MUT. Atmospheric pressure is measured using a digital barometer type PTB330 (manufacturer VAISALA Gmbh, Bonn, Germany).

Two volumetric master meters were used: (i) a wet drum meter type BSM-1 which cyclic volume is 5 dm^3^ (manufacturer BESSEL, Città di Castello, Italy), in the range from 0.04 to 0.6 m^3^/h; (ii) a G16 rotary gas meter type CGR-01 (manufacturer COMMON S.A., Łódź, Poland) in the range up to 6 m^3^/h. In [30] it was demonstrated that the diffusivity and solubility of hydrogen have no practical impact on the measurement with a wet drum gas meter. In fact, the diffusivity of hydrogen in water is comparable with that in air, and the measurement bias due to the diffusion of hydrogen in water with respect to air is 0.015% in the most stringent case. Additionally, in the present research, the use of oil instead of water reduces the solubility of hydrogen. To minimize the entrainment of oil by the gas stream and its consequent impact on the accuracy of the MUT, the reference wet gas meter was installed downstream to the MUTs during tests. Similarly, also for a rotary gas meter at high hydrogen contents a not negligible risk of underestimation at low flow rates occurs due to the unmeasured hydrogen flowing between the rotors and the gas meter body. However, in the present research, the rotary piston gas meter was custom-made for hydrogen measurements, with minimized internal leaks. It should also be noted that the measurements were performed with group H gas [24] and with a maximum admixture of 23%vol of hydrogen. Therefore, the hydrogen solubility in oil or internal leaks in the rotary gas meter do not significantly differ in respect to the measurements with natural gas. To validate the measurement method, cross measurements were carried out at the same flow rates using a drum wet and rotary gas meter, as well as by comparison with a mass flow meter, which allowed confirmation of the reliability of measurements.

Data acquisition is carried out via software and the following parameters were automatically gathered: (i) ambient conditions (i.e., barometric pressure, ambient temperature and humidity); (ii) gas pressure and temperature together with the volume pulses of the reference gas meters; (iii) gas pressure and temperature and pressure drop of the MUTs. The readings of the MUTs were taken manually from the meter display using, when available, the test mode resolution, depending on the gas meter type. Calibrated temperature and pressure transmitters were used at the test bench to measure thermodynamic conditions on the MUTs and reference gas meters, aiming at obtaining the respective actual reference volumes. Also, the corrections resulting from calibration are automatically applied by the software to get the volumes at reference conditions.

The three investigated G4 domestic gas meters (i.e., with $Q_{min}$ = 0.04 m^3^/h, $Q_{t}$ = 0.6 m^3^/h and $Q_{max}$ = 6 m^3^/h) were represented by a hybrid diaphragm gas meter, an ultrasonic gas meter and, finally, a thermal mass gas meter. These meters were all put into service between 2016 and 2017 in the distribution network of Mirano (in the district of Venice, northern Italy) and then removed in 2023. The experimental design was aimed at testing several possible distribution conditions, particularly: (i) test with air; (ii) test with natural gas of H family [24]; (iii) test with hydrogen admixtures (i.e., the same gas of H family with hydrogen content of 2, 5, 10 and 23%vol). In Table 2 the composition and main thermodynamic properties of the test gases have been summarized.

The error of indication of the meters was measured at five flow rate values (i.e., $Q_{min}$, $Q_{t}$, 0.4 $Q_{max}$, 0.7 $Q_{max}$ and $Q_{max}$). According to the provisions of the applicable standards, errors have been calculated in terms of volumes at standard condition. Therefore, flow rate is a test parameter, and it has been kept within ±5% of the nominal value by controlling the totalized volumes and test run duration. Each test cycle has been repeated three times and the average error has been calculated. The weighted mean error (WME) was also calculated according to [5] as per Equation (3), in which $k_{i}$ is the weighting factor and $E_{MUT,i}$ is the error of indication at the flow rate $Q_{i}$. The WME is not used for the purposes of the periodic verification and the relative MPE = 0.60% in the first verification is therefore only indicative. However, the WME value is very useful for providing an average performance of the meter.
(3)$$WME=\frac{\sum_{i}^{n}k_{i}\;E_{MUT,i}}{\sum_{i}^{n}k_{i}\;};\;\left\{\begin{array}{c} k_{i}=\frac{Q_{i}}{Q_{max}}\;\;for\;Q_{i}\leq 0.7Q_{max}\\ k_{i}=1.4-\frac{Q_{i}}{Q_{max}}\;\;for\;0.7Q_{max}<Q_{i}\leq Q_{max} \end{array}\right.$$

## 3. Results and Discussion

### 3.1. Diaphragm and Ultrasonic Gas Meters

In Table 3 and Table 4 the results of the tests of the diaphragm and ultrasonic gas meters have been reported in terms of error of indication at the different test flow rates (column 1) and test fluids (columns 3–8), together with the corresponding MPEs (column 2).

In Figure 2, the results of the verification of the diaphragm and the ultrasonic gas meters have been depicted graphically, together with the corresponding limits of the periodic verification (dotted lines). In Figure 3 the same results are reported highlighting the trends at different flow rates and test fluids.

It can be highlighted that the diaphragm and the ultrasonic gas meters passed the verification at all the flow rates and with all the test gases, including mixtures of natural gas and hydrogen. Both meters, in fact, comply even with the limits of the initial verification, which are stricter than the periodic in-service ones. However, from the analysis of the results, it can be further highlighted that:The WME of the diaphragm gas meter was found to be slightly above 0.6% for all test gases and errors of indication were all positive (i.e., in benefit of the DSO);The WME of the ultrasonic gas meter was found well below 0.60% and errors of indication were all negative (i.e., in benefit of the consumer), except in only two spare verification points.

Solely for the diaphragm gas meter, at $Q_{min}$ a decreasing trend was observed when the hydrogen content increases. This effect could be related to the higher volatility of the hydrogen in respect to the NG. The reliability of such types of gas meters with hydrogen admixture is also demonstrated by the narrow band of variation (i.e., within 0.4%) of the error of indication in the range $Q_{t}$ − $Q_{max}$ as the hydrogen admixtures vary. The fact that, for both meters the larger band of variation (i.e., within 0.6%, which is roughly the test uncertainty) has been observed at $Q_{min}$, further confirms the role of hydrogen volatility.

It can be therefore affirmed that diaphragm and ultrasonic gas meters are not significantly affected by the hydrogen admixture.

### 3.2. Thermal Mass Gas Meters

In Table 5 the results of the tests of the thermal mass gas meter in terms of error of indication at the different test flow rates (column 1) and test fluids (columns 3–8) have been reported, together with the corresponding MPE (column 2). Figure 4a graphically shows the errors of the thermal mass gas meter tested, together with the admitted limits (dotted lines). Finally, in Figure 4b the error trend as the flow rate and the test gas composition varies is depicted.

It can be observed that the outcome of the verification of the thermal gas meter was positive for the test in air, gas of H family and with 2%vol of hydrogen admixture. At higher hydrogen content (i.e., 5, 10 and 23%vol) the outcome was negative, with errors of indication largely exceeding the corresponding MPEs, especially with the admixture of 10%vol of hydrogen. For the sake of truth, in the Italian NG grid, a limit of 2%vol of hydrogen in the NG is currently allowed [31] and the meter tested relies on a first-generation sensor, which has not been designed and approved for complying with high hydrogen contents. Therefore, it can be affirmed that the investigated thermal mass gas meter fulfils the current regulation. Nevertheless, errors with air were all negative (i.e., in benefit of the consumer) whereas those with 2%vol of hydrogen admixture were all positive (i.e., in benefit of the DSO).

From the analysis of the measured data, it seems that a 2%vol hydrogen admixture represents a limit for the investigated thermal mass gas meter, since for higher hydrogen admixtures the error curve first tends to rise (i.e., at 5 and 10%vol) and then to shift towards negative values (i.e., 23%vol). Furthermore, the following evidence also emerged: (i) the progressive increase of the error magnitude as the hydrogen content increases; (ii) the decrease of the errors of indication toward negative values as the flow rate decreases, especially at higher hydrogen admixture (i.e., 10 and 23%vol). Also in this case, the latter finding could be related to the higher volatility of hydrogen in respect to the natural gas.

Finally, the gas–air relationship of the three domestic gas meters investigated has been reported in Table 6 in terms of difference between errors of indication with gas and air at different test flow rates (column 1), together with the corresponding MPE (column 2). For the sake of synthesis, the results of tests with gas of H family have only been reported. In any case, the admitted limit established by clause 5.4 of EN 17526 was never exceeded, even with hydrogen admixtures. 

It is worth noting that the results obtained in the present research are consistent with those of other studies available in the literature and, particularly, with [14,15,23] for diaphragm gas meters, [9,30] for ultrasonic and [16] for thermal mass.

In particular, ultrasonic gas meters seem to be almost insensitive to hydrogen admixtures. Diaphragm gas meters, which rely on a volumetric measuring principle, are influenced only at minimum flow rate, due to the higher hydrogen volatility combined with larger flowing time of the gas in the measuring chambers. On the other hand, the increase of the speed of sound (as well as the decrease of density) as the hydrogen admixture increases leads to limited effects on the accuracy of domestic ultrasonic gas meters. In fact, the measuring principle itself (i.e., the measurement of the sing-around time of flight in one direction first and thereafter in the opposite direction of the flow) theoretically eliminates the influence of the speed of sound in the gas stream. The smooth changes observed in the accuracy due to hydrogen admixtures are thus attributable to the very limited changes in the time-of-flight measurements.

Domestic thermal mass gas meters are instead potentially more affected by hydrogen admixture. In fact, the change of the dynamic viscosity and consequently of the Reynolds number can significantly alter the ratio of the main flow rate to that in the capillary, leading to potentially high inaccuracy. This is why manufacturers have recently developed a second generation of this sensor, relying on specific design routines capable of addressing changes of the gas properties by applying appropriate correction factors [32]. In any case, the currently installed thermal mass sensors of the first generation would be capable of addressing a hydrogen admixture up to 2%vol.

## 4. Conclusions

In this paper, the reliability of in-service domestic gas meters has been investigated by performing an experimental campaign in the laboratory on three G4 gas meters with different flow sensors removed from the field. Tests were performed with air, gas of H family and with hydrogen admixtures up to 23%vol, with the aim of simulating gas compositions that can become realistic in the immediate future.

The obtained results show that both diaphragm and ultrasonic gas meters seem to be not particularly affected by hydrogen admixture, since the outcome of the tests was positive at all verification points and with all test gases. Indeed, a contrasting behavior of the tested thermal gas meter (which relies on a first-generation sensor) emerges. In particular, verification was positive for the test in air, gas of H family and with hydrogen content of 2%vol, which is the current admitted limit in the Italian NG grid. It can be therefore affirmed that the tested thermal gas meter is compliant with the current regulation. On the other hand, reliability with hydrogen content above 2%vol was not demonstrated and, in particular:The outcome of the verification was negative, showing high errors, especially with hydrogen content of 10%vol;A progressive increase of the absolute errors (i.e., weighted mean error up to about 15.8%) as the hydrogen content increases was highlighted.

The results of the present research should be useful for setting appropriate provisions for the update of technical standard and national regulations and for the development of guidelines for subsequent verification of gas meters. Furthermore, given the current prospects for the diffusion of hydrogen blending, DSO could base their development plans on the basis of the domestic gas meters installed, in order to best protect consumers and guarantee the optimal balancing of the grid.

## Figures and Tables

**Figure 1 sensors-24-01455-f001:**
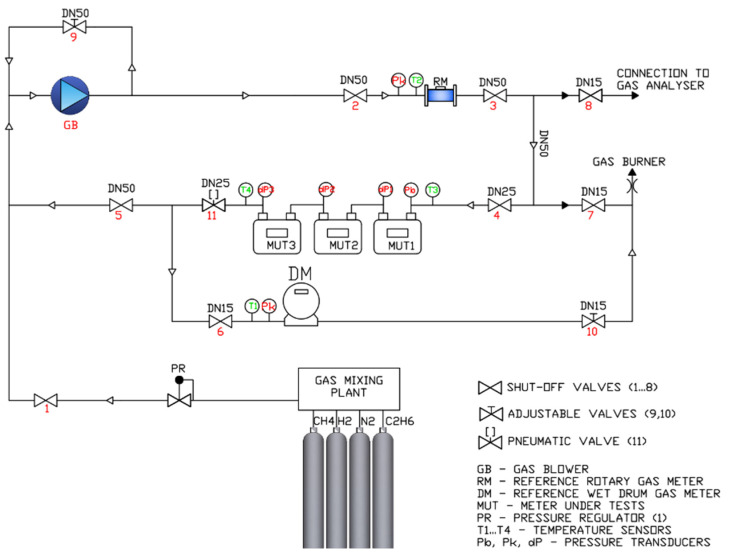
Sketch of the test bench of the Oil and Gas National Research Institute (PL).

**Figure 2 sensors-24-01455-f002:**
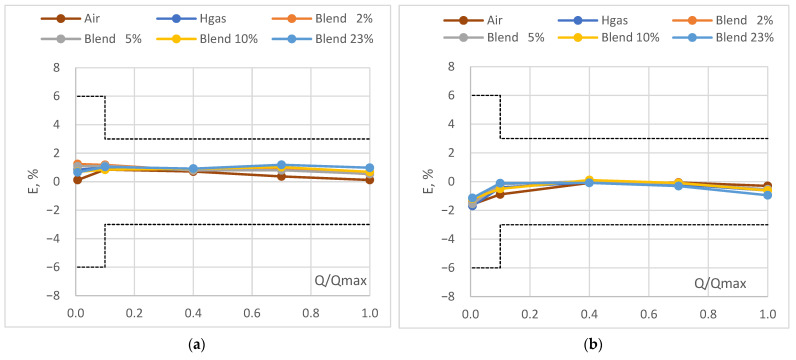
Error curve: (**a**) diaphragm gas meter, (**b**) ultrasonic gas meter.

**Figure 3 sensors-24-01455-f003:**
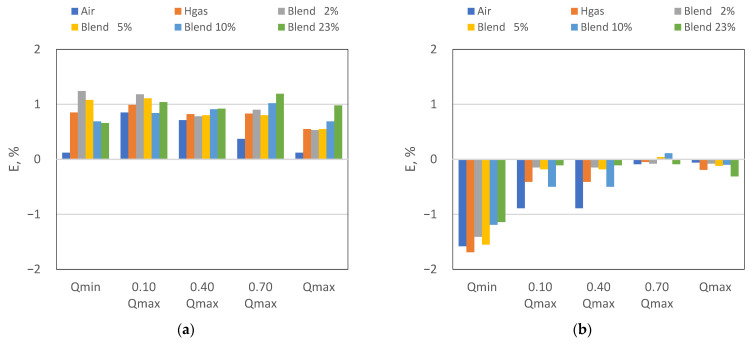
Error trend: (**a**) diaphragm gas meter, (**b**) ultrasonic gas meter.

**Figure 4 sensors-24-01455-f004:**
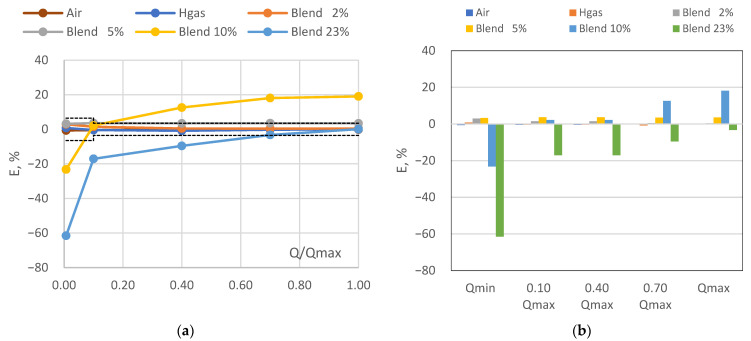
Thermal mass gas meter: (**a**) error curve; (**b**) error trend.

**Table 1 sensors-24-01455-t001:** Maximum permissible errors (MPE) in subsequent verification for gas meters.

Flow Rate	Class 1	Class 1,5	Class 1,5 *
$Q_{min}\;\leq \;Q<Q_{t}$	±4%	±6%	±7%
$Q_{t}\;\leq \;Q<Q_{max}$	±2%	±3%	±4%

* with conversion device compensating the gas temperature.

**Table 2 sensors-24-01455-t002:** Characteristics of the test fluid used in the experimental campaign (T = 15 °C, P = 1.01352 bar).

	Air	Hgas	Blend 2% H_2_	Blend 5% H_2_	Blend 10% H_2_	Blend 23% H_2_
Methane, CH_4_	-	0.910	0.892	0.865	0.819	0.701
Ethane, C_2_H_6_	-	0.050	0.049	0.048	0.045	0.039
Nitrogen, N_2_	-	0.040	0.039	0.038	0.036	0.031
Hydrogen, H_2_	-	0.000	0.020	0.050	0.100	0.230
Relative density (-)	-	0.596	0.585	0.569	0.543	0.474
Density (kg m^−3^)	1.225	0.728	0.715	0.695	0.663	0.579
Gross calorific value (MJ Sm^−3^)	-	37.696	37.181	36.409	35.123	31.785
Wobbe Index (MJ Sm^−3^)	-	48.841	48.607	48.257	47.671	46.155
Specific heat (J kg^−1^ K^−1^)	1011.0	2142.2	2171.1	2216.4	2297.8	2551.8
${\rho \;\bar{c}}_{p}$ (kJ m^−3^/K^−1^)	1.238	1.559	1.552	1.541	1.524	1.478

**Table 3 sensors-24-01455-t003:** Measured errors of indication of the diaphragm gas meter (%).

Flow Rate	MPE	Air	Hgas	Blend 2% H_2_	Blend 5% H_2_	Blend 10% H_2_	Blend 23% H_2_
$Q_{min}$	±6.00	0.12	0.85	1.24	1.08	0.69	0.66
$Q_{t}$	±3.00	0.85	0.99	1.18	1.11	0.84	1.04
0.4 $Q_{max}$	±3.00	0.71	0.82	0.78	0.80	0.91	0.92
0.7 $Q_{max}$	±3.00	0.37	0.83	0.90	0.80	1.02	1.19
$Q_{max}$	±3.00	0.12	0.55	0.53	0.55	0.69	0.98
WME	-	0.42	0.77	0.80	0.76	0.90	1.06
Outcome	-	Pass	Pass	Pass	Pass	Pass	Pass

**Table 4 sensors-24-01455-t004:** Measured errors of indication of the ultrasonic gas meter (%).

Flow Rate	MPE	Air	Hgas	Blend 2% H_2_	Blend 5% H_2_	Blend 10% H_2_	Blend 23% H_2_
$Q_{min}$	±6.00	−1.58	−1.69	−1.41	−1.55	−1.19	−1.14
$Q_{t}$	±3.00	−0.89	−0.41	−0.15	−0.18	−0.50	−0.11
0.4 $Q_{max}$	±3.00	−0.09	−0.05	−0.08	0.04	0.11	−0.09
0.7 $Q_{max}$	±3.00	−0.06	−0.19	−0.08	−0.12	−0.10	−0.31
$Q_{max}$	±3.00	−0.30	−0.62	−0.62	−0.54	−0.62	−0.95
WME	-	−0.18	−0.28	−0.22	−0.19	−0.21	−0.40
Outcome	-	Pass	Pass	Pass	Pass	Pass	Pass

**Table 5 sensors-24-01455-t005:** Measured errors of indication of the thermal mass gas meter (%).

Flow Rate	MPE	Air	Hgas	Blend 2% H_2_	Blend 5% H_2_	Blend 10% H_2_	Blend 23% H_2_
$Q_{min}$	±7.00	−0.63	0.81	2.94	3.24	−23.20	−61.52
$Q_{t}$	±4.00	−0.46	−0.42	1.50	3.62	2.21	−17.08
0.4 $Q_{max}$	±4.00	−0.14	−0.81	0.49	3.46	12.61	−9.57
0.7 $Q_{max}$	±4.00	−0.12	−0.24	0.39	3.55	18.11	−3.24
$Q_{max}$	±4.00	−0.11	0.03	0.46	3.47	19.05	0.13
WME	-	−0.15	−0.32	0.51	3.50	15.78	−5.07
Outcome	-	Pass	Pass	Pass	Fail	Fail	Fail

**Table 6 sensors-24-01455-t006:** Gas–air relationship (%).

Flow Rate	MPE	Diaphragm	Ultrasonic	Thermal Mass
$Q_{min}$	±3.0	0.73	−0.11	1.44
$Q_{t}$	±1.5	0.14	0.48	0.04
0.4 $Q_{max}$	±1.5	0.14	0.48	0.04
0.7 $Q_{max}$	±1.5	0.11	0.04	−0.67
$Q_{max}$	±1.5	0.46	−0.13	−0.12

## Data Availability

Data available on request.

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
