# Peer review of "Reliability of Domestic Gas Flow Sensors with Hydrogen Admixtures"

_sensors, 2024, doi:10.3390/s24051455_

Round 1

Reviewer 1 Report

Comments and Suggestions for Authors

Comments and Suggestions for Authors

Review Comments for Manuscript Number: sensors-2825036-peer-review-v1

Title:

Reliability of domestic gas flow sensors with hydrogen admixtures

Journal:

Sensors MDPI

Domestic gas smart meters based on static measuring sensors are discussed in the current research. In this study, micro electro-mechanical sensors and ultrasonic time of flight sensors have been used. Due to gas composition in networks and hydrogen admixture, these sensors may be affected in the future. The purpose of this work was to investigate the reliability of smart gas meters in service. There appear to be no particular issues with diaphragm and ultrasonic gas meters, however thermal mass gas meters exceed the limits for subsequent verification when hydrogen content exceeds 2% vol. Here are the comments on tis work:

1.      The work looks fine but I don’t see novelty in it. Add the contribution of this paper clearly at the end of the introduction “The novelty of this paper is represented by the fact that the effect of the hydrogen injection in the NG on installed domestic gas meters of different measuring principle is a topic not widely investigated in the scientific literature and applicable standard documents, as well as the potential deviation between subsequent verification outcomes when tests are performed in air and in gas.”

2.      The abstract must be revised carefully. More results should be presented.

3.      Major English revision is needed. Try to make your sentences shorter; not four lines sentence. It will help the readers to follow up.

4.      The figures must be replaced with highly resolution ones.

5.      In which region of Italy gave you conducted this work?

6.      What is the accuracy of the used measurements equipment?

7.      Make a comparison with other works if found.

8.      Check your references such as the one in line 305, … etc.

9.      More discussion should be added, especially for the tables.

I advise the authors to go through prior research in order to support their work.

Comments on the Quality of English Language

Moderate editing of English language required

Reviewer 2 Report

Comments and Suggestions for Authors

Comments on the Quality of English Language

Reviewer 3 Report

Comments and Suggestions for Authors

The author studied the reliability of sensors, especially H contents. Since Hydrogen is functionalizing in various fields of engineering, the research will be valuable for readers.

Overall, the experiment section is weak to provide new findings and novelty of the paper.

Author must add more explanations about plots and tables. Readers cannot accept author's discussion without details.

Author can provide model names of each sensor and method of data acquisition processes. Be clear of no noise and structural error for someone who may reproduce the finding.

For the figure1 schematic, the author provides the function of every part.

The author must replace many 'spread' 'spreading' to other expressions.

The author missed the full name of OIML.

Fig 1 is invisible.

What is the dash line of Figure 2?

Remove repeated 'it is worth nothing that'.

The author should consider changing the type of plots in Figures 3, 4b, and 5.

Remove decimals in error plots.

Round 2

Reviewer 1 Report

Comments and Suggestions for Authors

The authors have improved their manuscript in accordance with the reviewer's instructions.

Comments on the Quality of English Language

Minor editing of English language required

Author Response

We thank the reviewer for his positive judgment.

Reviewer 2 Report

Comments and Suggestions for Authors

Thank you for the revised version and implementing the comments. 

Comments on the Quality of English Language

Some editing of English may still be needed. 

Author Response

(The authors gave the same response as above.)

Reviewer 3 Report

Comments and Suggestions for Authors

Improve the author's response to comment 2. There are a lot of data in the tables. The author must kindly explain the significant of each piece of data.

Improve the author's response to comment 4. The author must kindly explain each part and the systematic flow of the schematic. 

Improve analytical explanations in every plot. Overall, analytical explanations for figures are too short. The author must kindly explain data to readers in every table and plot. Please note that if the author inserts figures and tables, the author must take responsibility for the reader's understanding through kind explanations.
